# ABPT: Amended Backpropagation through Time with Partially Differentiable Rewards

## Abstract

Quadrotor control policies can be trained with high performance using the exact gradients of the differentiable rewards to optimize policy parameters via backpropagation-through-time (BPTT). However, designing a fully differentiable reward architecture is often challenging in real-world high-level tasks rather than control in simulation. Partially differentiable rewards will result in biased gradient propagation that severely degrades training performance. To overcome this limitation, we propose Amended Backpropagation-through-Time (ABPT), a novel approach that mitigates gradient bias while preserving the training efficiency of BPTT. ABPT combines learned 0-step returns and analytical cumulative rewards, effectively reducing the bias by leveraging value gradients from the learned Q-value function. Additionally, it adopts entropy regularization and state initialization mechanisms to improve training efficiency. We evaluate ABPT on four representative quadrotor flight tasks in both real world and simulation. Experimental results demonstrate that ABPT converges significantly faster and achieves higher ultimate rewards than existing representative learning algorithms, particularly in tasks involving partially differentiable rewards.

## 1 Introduction

Quadrotors have demonstrated significant potential in various real-world applications including wild rescue, dangerous high-altitude work, and delivery. Recent work (Loquercio et al. (2021; 2019); Kaufmann et al. (2018)) has shown end-to-end policies can be learned through imitation learning for controlling quadrotors from raw sensory data. However, the performance is largely restricted by expert's capability. Though reinforcement learning (RL) can address this limitation through self-exploration, its policy updates rely on gradient approximations (Sutton & Barto (2018)), which require extensive sampling or replay mechanisms and often result in slow convergence and suboptimal training outcomes. Compared with imitation learning and traditional RL algorithms, recent studies (Zhang et al. (2024); Wiedemann et al. (2023); Lv et al. (2023); Song et al. (2024); Hu et al. (2025)) have demonstrated that directly leveraging first-order gradients for policy learning leads to faster convergence and superior performance, particularly in quadrotor tasks (Zhang et al. (2024); Wiedemann et al. (2023)).

Using first-order gradients for training requires not only the dynamics but also the reward function to be differentiable. However, designing fully differentiable rewards is often impractical for complex quadrotor tasks. Reward functions in such scenarios often include non-differentiable components, such as conditional constants or binary scores (e.g., granting points upon gate crossing in a racing task or upon object detection in a search task), which violate differentiability requirements. These non-differentiable elements disrupt the computation graph during backpropagation-through-time (BPTT), leading to biased first-order gradients—a phenomenon we term **Biased Gradient**. This bias misguides training, causing optimization to stall in local minima and deviate from the intended direction of improvement.

To address this issue in quadrotor tasks, we propose an on-policy actor-critic approach - **Amended Backpropagation-through-Time (ABPT)**, which mitigates the bias gradient introduced by the non-differentiable rewards while keeping high policy learning performance in terms of training speed and converged rewards. Our approach combines 0-step returns with N-step returns (Sutton & Barto (2018)), leveraging value gradients generated by the 0-step returns to balance first-order gradient

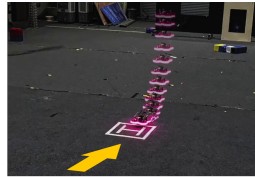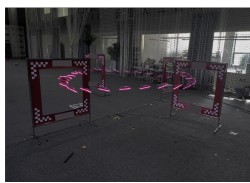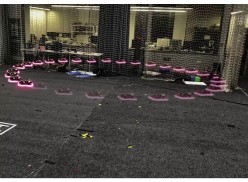

Figure 1: Our trained policies were deployed in the real world with zero-shot sim-to-real transfer. Additional results are provided in the supplementary video, which showcases four tasks : hovering, landing, racing, and tracking, from left to right.

accuracy and exploitation. Additionally, ABPT incorporates entropy to suppress negative impact by the instability of critic learning. It also employs a replay buffer to store state experiences, initializing episodes with these states to enhance sampling efficiency. We evaluate our method on four representative quadrotor tasks, comparing it against classic policy gradient and first-order gradient methods. These tasks are designed to progressively increase the reward non-differentiability, testing the adaptability of each approach. Experimental results demonstrate that ABPT achieves the fastest convergence and highest final rewards across all baselines. This superiority is attributed to ABPT's ability to compensate for biased gradients and enhance exploration via entropy regularization and state replay. Furthermore, ABPT exhibits robustness across varying learning rates and reward structures. Our technical contributions are summarized as follows:

- We propose ABPT, a novel approach to address the challenges in first-order gradient learning, including biased gradients caused by non-differentiable rewards and susceptibility to local minima.

- We provide a comprehensive analysis of ABPT's effectiveness, offering insights to advance differentiable physics-based learning methods.

- We validate ABPT-trained policies of four representative quadrotor tasks in the real world through zero-shot sim-to-real transfer.

## 2  RELATED WORK

### 2.1  REINFORCEMENT LEARNING

Traditional reinforcement learning can be divided into two classes: model-free RL and model-based RL. Model-free RL includes value-based and policy-gradient methods. Value-based methods learn value functions to estimate long-term rewards. DQN (Mnih (2013)) introduced neural networks for discrete actions, while DDPG (Lillicrap (2015)) extended this to continuous action spaces. TD3 (Fujimoto et al. (2018)) reduced overestimation bias with multiple value networks, and SAC (Haarnoja et al. (2018)) used a maximum entropy framework for robust high-dimensional learning. Policy-gradient methods directly optimize policies using gradients. TRPO (Schulman (2015)) stabilized updates via trust regions, and PPO (Schulman et al. (2017)) simplified optimization with a clipped surrogate objective.

In contrast to model-free RL which treats the environment as a black box, model-based RL (Moerland et al. (2023)) introduces an additional process to learn the environment's dynamics. For example, PILCO (Deisenroth & Rasmussen (2011)) and Dyna-Q (Sutton (1990)) leveraged learned environment models to generate simulated experiences to accelerate training. Methods like (Chua et al. (2018); Watter et al. (2015)) employ trajectory sampling to plan over learned environment models. Dreamer (Hafner et al. (2019)) embedded entire functions into a latent space, enabling end-to-end policy updates via backpropagation-through-time (BPTT). Despite their advantages, existing RL methods do not explicitly utilize the dynamics of robotics that can be precisely described by physical laws.

## 2.2 Differentiable Simulators

Policy learning via differentiable physics is an approach that integrates the physical simulations with differentiable dynamics to enable policy learning directly by using gradient-based optimization. Making the dynamics differentiable in the simulator is the key to this approach. DiffTaichi (Hu et al. (2020)) is a comprehensive differentiable physics engine that includes simulations of fluid, gas, rigid body movement, and more. In the field of robotics, Brax (Freeman et al. (2021)) offers differentiable versions of common RL benchmarks, built on four physics engines, including JAX and MuJoCo (Todorov et al. (2012)). Another line of research focuses on addressing challenges in contact-rich environments. For example, Heiden et al. (Heiden et al. (2021)) tackle the contact-rich discontinuity problem in quadruped robots by employing a neural network to approximate the residuals. Dojo (Howell et al. (2023)) enhances contact solvers and integrates various integrators to accelerate computations while maintaining fidelity. VisFly (Li et al. (2024)) introduces a versatile drone simulator with fast rendering, based on Habitat-Sim (Savva et al. (2019)), providing a platform for high-level applications. To enhance the efficiency, many simulators leverage GPU-accelerated frameworks like JAX (Schoenholz & Cubuk (2020)) and PyTorch (Paszke et al. (2017)) for faster computations.

## 2.3 First-order Gradient Training

With the differentiable simulators, the policy can be trained through BPTT by using the first-order gradients. Though first-order gradients enable faster and more accurate gradient computation, they suffer from gradient explosion/vanishing or instability caused by smooth dynamics. Many attempts have tried to address these issues and strengthen robustness. PODS (Mora et al. (2021)) leverages both first- and second-order gradients with respect to cumulative rewards. SHAC (Xu et al. (2022)) employs an actor-critic framework, truncates the learning window to avoid vanishing/exploding gradients, and smooths the gradient updates. AOBG (Suh et al. (2022)) combines ZOG (policy gradient) with FOG, using an adaptive ratio based on gradient variance in the minibatch to avoid the high variance typical of pure FOG in discontinuous dynamics. AGPO (Gao et al. (2024)) replaces ZOG in mixture with critic predictions, as Q-values offer lower empirical variance during policy rollouts. While both AGPO and AOBG converge to asymptotic rewards in significantly fewer timesteps, the mixture ratio requires excessive computational resources, leading to longer wall-time. AHAC (Georgiev et al. (2024)) makes the horizon adaptive to reduce sampling error in scenarios involving stiff dynamics. SAPO (Xing et al. (2024)) introduces entropy to strengthen the training stability especially in soft-body simulation. All these variants are designed to improve training efficiency and have been validated on controlled simulation benchmarks. However, although SHAC, SAPO, and AHAC incorporate critics for learning, their value functions are positioned only at the end of the horizon, which prevents them from addressing the gradient bias introduced by non-differentiable rewards within the horizon (as explained in Section 4).

## 3 Preliminaries

The goal of reinforcement learning is to find a stochastic policy $\pi$ that maximizes the expected cumulative reward, or the expected return, over a trajectory $\tau$. In a common actor-critic pipeline, both the actor $\pi_\theta$ and the critic – either the action-value function $Q_\phi(s, a)$ or the state-value function $V_\phi(s) = \mathbb{E}_{a \sim \pi_\theta}[Q_\phi(s, a)]$ – are approximated by neural networks with parameters $\theta$ and $\phi$. The key problem is how to estimate the gradients to optimize the expected return. The methods could be divided into two following categories:

**Policy Gradient.** Policy gradient methods estimate the gradient of the expected return using the log-probability of sample trajectories, conditioned on the policy's action distribution. Given a batch of experience, the policy gradient is computed as:

$$\nabla_\theta^{[0]} \mathcal{J}_\theta = \frac{1}{|\mathcal{B}|} \left[ \sum_{\tau \in \mathcal{B}} \sum_{t=0}^{T} \nabla_\theta \log \pi_\theta(a_t \mid s_t) A^{\pi_\theta}(s_t, a_t) \right], \tag{1}$$

where $A^{\pi_\theta}(\cdot)$ represents the advantage derived from the value functions using current policy, $\mathcal{B}$ denotes the minibatch of sampled trajectories, $\tau$ represents a trajectory within the minibatch. Because

this formulation does not require differentiating through the environment dynamics, it is also named zeroth-order gradient (ZOG).

**Value Gradient.** Value gradient methods compute the policy gradient by differentiating through the action-value function:

$$\nabla_\theta^{[q]} \mathcal{J}_\theta = \frac{1}{|\mathcal{B}|} \left[ \sum_{i=1}^{|\mathcal{B}|} \nabla_\theta Q_\phi\big(s^i, \pi_\theta(s^i)\big) \right] \tag{2}$$

(Gao et al. (2024)) named this gradient estimator as Q gradient (QG). Compared with ZOG, the accuracy of value-function approximation is particularly critical for actor training, since QG relies directly on backpropagation through the action-value function. In contrast, ZOG estimates advantages with respect to the current policy, which makes actor training more robust to imperfections in critic learning.

## 4 FIRST-ORDER GRADIENT APPROACH WITH NON-DIFFERENTIABLE REWARDS

**First-order Gradient.** Given the state dynamics $T$ and reward function $R$ being differentiable, one can compute the exact gradients of the expected return for policy learning via backpropagation through time. This exact gradient estimate is called first-order gradient (FOG):

$$\nabla_\theta \mathcal{J}_\theta = \left( \sum_{k=0}^{N-1} \gamma^k \frac{\partial R(s_{t+k})}{\partial \theta} \right), \tag{3}$$

where $N$ represents the horizon length, $i$ denotes the $i$-th trajectory within the minibatch, and $R$ represents the reward function. To consider infinite return while avoiding gradient explosion, an approximated N-step return (Sutton & Barto (2018)) has been introduced in (Xu et al. (2022)):

$$\nabla_\theta \mathcal{J}_\theta = \left( \sum_{k=0}^{N-1} \gamma^k \frac{\partial R(s_{t+k})}{\partial \theta} \right) + \gamma^N \nabla_\theta V_\phi(s_{t+N}). \tag{4}$$

Here, $V_\phi$ is the state-value function reparameterized by $\phi$. As shown in (Xu et al. (2022)), using this approximated N-step return can introduce smooth landscape for optimization and mitigate the gradient explosion issues. However, it cannot address non-differentiable rewards as we will discuss later. Compared to Equation (2) and Equation (1), Equation (4) incorporates component that could be optimized by precise gradient descent.

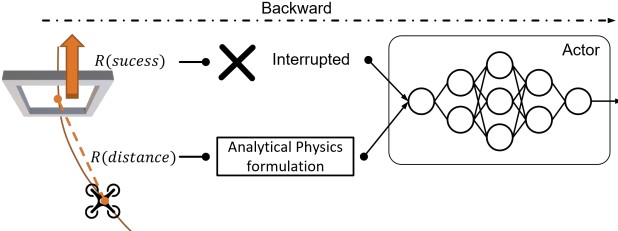

Figure 2: An illustration for explaining biased gradient. In a racing task introduced in Section 6.1, the reward for passing the gate is a conditional constant, unable to automatically compute gradients.

**Biased Gradient.** When the rewards are partially differentiable, the gradients of non-differentiable part of the rewards will be absent from backpropagation. For example, as shown in Figure 2, a racing task's reward function consists of two components. The first one $R_{dist}$ depends on the distance from the drone to the gate to encourage the drone to move toward the gate, which is differentiable w.r.t the state. The second one $R_{succ}$ is a conditional constant score given for successfully passing the gate, which does not involve gradient computation w.r.t. policy parameters. Therefore, although the desired objective involves both rewards

$$\mathcal{J}_\theta = \left( \sum_{k=0}^{N-1} \gamma^k \Big( R_{dist}(s_{t+k}) + R_{succ}(s_{t+k}) \Big) \right) + \gamma^N V_\phi(s_{t+N}), \tag{5}$$

backpropagation-through-time can effectively optimize only the differentiable components:

$$\mathcal{J}_\theta = \left( \sum_{k=0}^{N-1} \gamma^k R_{dist}(s_{t+k}) \right) + \gamma^N V_\phi(s_{t+N}), \tag{6}$$

As a result, the gate crossing reward $R_{succ}$, despite being crucial for learning the expected behavior (e.g. crossing the gate), is ignored during training. This ignorance can hinder the learned policy's ability to perform the desired actions.

## 5 THE PROPOSED METHOD

As previously discussed, explicit use of first-order gradients for policy learning requires addressing gradient bias caused by non-differentiable rewards. Motivated by the value gradient method, we propose to combine the 0-step return with N-step return for policy learning. This combination mitigates the gradient bias while leveraging the strength of both gradient types. Our method, **A**mended **B**ackpropagation-through-**T**ime (ABPT), is an on-policy actor-critic learning approach. An overview is presented in Figure 3.

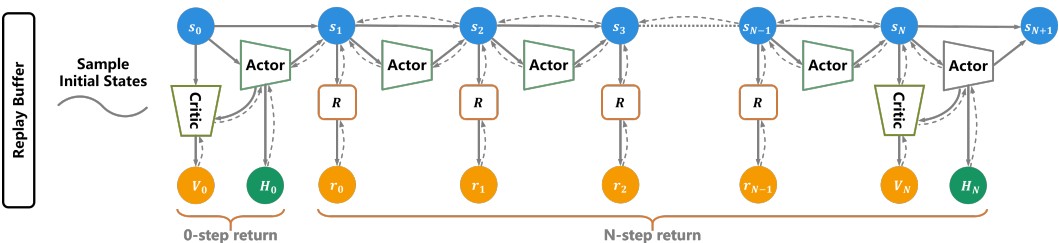

Figure 3: An overview of ABPT. ABPT combines 0-step return and N-step return together, to compensate the biased gradient resulting from partially non-differentiable reward. The red dash lines indicate the direction of backpropagation. The replay buffer stores only visited states for episode initialization to improve sampling efficiency, irrelevant to training.

During each training episode, we collect $|\mathcal{B}|$ trajectories with a horizon length $N$ and optimize the following objective function to update the actor network parameters $\theta$:

$$\mathcal{J}_\theta = \frac{1}{2|\mathcal{B}|} \sum_{i=1}^{|\mathcal{B}|} \left( \mathcal{J}_\theta^N + \mathcal{J}_\theta^0 \right) \tag{7}$$

where $\mathcal{J}_\theta^N$, $\mathcal{J}_\theta^0$ are N-step return and 0-step return, defined as

$$\mathcal{J}_\theta^N = \underbrace{\left( \sum_{k=0}^{N-1} \gamma^k R(s_{t+k}^i) \right)}_{\mathcal{G}_\theta^{t:t+N}} + (1-d)\gamma^N \underbrace{V_\phi(s_{t+N}^i)}_{\mathcal{V}_{\theta|\phi}^{t+N+1}}, \qquad \mathcal{J}_\theta^0 = \underbrace{V_\phi(s_t^i)}_{\mathcal{V}_{\theta|\phi}^t} \tag{8}$$

Here, $d$ is a boolean variable indicating whether the current episode has ended, and $i$ denotes the trajectory index. Because each trajectory is generated by $\pi_\theta$, all terms are differentiable with respect to $\theta$. $\mathcal{G}_\theta^{t:t+N}$ represents the accumulated reward within the horizon and $\mathcal{V}_{\theta|\phi}^{t+N+1}$ is the value obtained by fixed critic. Both 0-step return and N-step return are expected values computed from the same action-value function $Q_\phi$. Ideally, if the critic $Q_\phi$ is learned perfectly, we have $J_\theta = J_\theta^N = J_\theta^0$. We prove using the objective function (7) for gradient computation is equivalent to combining both the value gradient and the first-order gradient for backpropagation in Appendix A.

We use a Gaussian policy $\pi_\theta(a|s) = \mathcal{N}(\mu_\theta(s), \sigma_\theta(s))$ for the actor network and apply the reparameterization trick (Kingma (2013)) to gradient computation. We also normalize the actions using tanh function to stabilize the training process: $a_t = \tanh(\mu_\theta(s_t) + \sigma_\theta(s_t)\epsilon)$, where $\epsilon \sim \mathcal{N}(0, I)$. After

---

**Algorithm 1** The proposed ABPT algorithm

---

1: Initialize parameters $\phi, \phi^-, \theta$ randomly, initialize state buffer $\mathcal{D} = \{\}$.
2: **while** num time-steps < total time-steps **do**
3:      # *Evaluate and collect states*
4:      **for** collecting steps = $1 \ldots i$ **do**
5:          Add states $\mathcal{D} \leftarrow \mathcal{D} \cup \{(s_i)_{i=1}^N\}$
6:      **end for**
7:
8:      # *Train actor net*
9:      Sample minibatch $\{(s_i)\}_{\mathcal{B}} \sim \mathcal{D}$ as initial states
10:      Compute the gradient of $\mathcal{J}_\theta$ and update the actor by gradient ascent $\theta \leftarrow \theta + \alpha \nabla_\theta \mathcal{J}_\theta$
11:
12:      # *Train critic net*
13:      Compute the estimated value $\tilde{V}_\phi$ using (10)
14:      **for** critic update step $c = 1..C$ **do**
15:          Compute the gradient of $\mathcal{L}_\phi$ and update weights by gradient descent $\phi \leftarrow \phi - \alpha \nabla_\phi \mathcal{L}_\phi$
16:          Softly update target critic $\phi^- \leftarrow (1 - \tau)\phi^- + \tau\phi$
17:      **end for**
18: **end while**

---

updating the critic, target returns are estimated over time and used to further refine the critic network parameters $\phi$ by minimizing the MSE loss function:

$$\mathcal{L}_\phi = \mathbb{E}_{s \in \{\tau_i\}} \left\| V_\phi(s) - \tilde{V}_\phi(s) \right\|^2 . \tag{9}$$

We employ TD($\lambda$) formulation (Sutton & Barto (2018)) to estimate the expected return using exponentially averaging $k-$step returns:

$$\tilde{V}_\phi(s_t) = (1 - \lambda) \left( \sum_{k=1}^{N-t-1} \lambda^{k-1} G_t^k \right) + \lambda^{N-t-1} G_t^{N-t} \tag{10}$$

where $G_t^k$ denotes $k-$step return from $t$:

$$G_t^k = \left( \sum_{l=0}^{k-1} \gamma^l r_{t+l} \right) + (1-d)\gamma^k V_\phi(s_{t+k}). \tag{11}$$

where $d \in \{0, 1\}$ indicates task termination. The state-value function is derived from the action-value function:

$$V_\phi(s) = \mathbb{E}_{a \sim \pi} [Q_\phi(s, a)] + \kappa H(\pi_\theta(\cdot \mid s)), \tag{12}$$

where we adopt an extra policy entropy term $H(\pi_\theta(\cdot \mid s))$ to encourage exploration as in SAC (Haarnoja et al. (2018)). $\kappa$ is an adaptive ratio whose computation follows (Haarnoja et al. (2018)). To stabilize the critic training, we follow (Mnih et al. (2015)) to use a target critic $\phi^-$ to estimate the expected return (see Equation (10)).

Existing methods (Xu et al. (2022)) start each new horizon at the end of the previous horizon, which prevents certain regions of the state space from serving as initial states, resulting in inefficient sampling (see Appendix D). To further encourage broader exploration during policy learning, we adopt a replay buffer to store all visited states throughout training. This buffer enables random sampling of dynamically feasible states for episode initialization. While conceptually similar to the replay buffer used in off-policy learning algorithms, our approach differs in that we store only visited states rather than transitions, and use these states solely for initialization, preserving the on-policy nature of training. The pseudo code of the proposed method is shown in Algorithm 1.

# 6 EXPERIMENTS

We address the following questions in this section: 1) How does ABPT improve performance on typical quadrotor tasks compared to baseline methods? 2) What distinctive advantages does ABPT exhibit in behavior? 3) What is the contribution of each individual component?

## 6.1 EXPERIMENT SETUP

We conduct the evaluation on four quadrotor tasks, hovering, tracking, landing, racing, which involve different levels of complexity. The hovering and tracking employ purely differentiable rewards. In contrast, both the landing and racing tasks incorporate binary rewards. However, there is a key difference between them. In landing, the continuous reward teaches the quadrotor to gradually slow down and descend, while the binary reward serves only to confirm successful touchdown. In racing, however, the binary reward plays a decisive role by preventing the quadrotor from hovering near the gates without actually passing through them.

In our experiments, we evaluate the proposed ABPT against three widely used baseline methods: PPO (Schulman et al. (2017)), BPTT (Freeman et al. (2021)), and SHAC (Xu et al. (2022)). PPO and SAC (Haarnoja et al. (2018)) remain among the most popular model-free algorithms for policy training, due to their stability and robustness to hyperparameters. However, SAC is not included in our comparisons because, in high-dimensional observation spaces, the critic requires substantially longer training time, making it less competitive (see Appendix C.4). Among first-order-gradient-based methods, SHAC is considered the most suitable baseline, as other approaches either exhibit slower wall-time training or share similar features with SHAC.

## 6.2 BENCHMARK TASKS

**Hovering.** Starting from a random position, the quadrotor needs to hover stably at a target location. Fully differentiable rewards are used in this task.

**Tracking.** Starting from a random position, the quadrotor tracks a circular trajectory with a fixed linear velocity. Fully differentiable rewards are used in this task.

**Landing.** Starting from a random position, the quadrotor gradually descends, and eventually lands at the required position on the ground. This task involves using non-differentiable rewards during training.

**Racing.** The quadrotor flies through four static gates as quickly as possible in a given order repeatedly. This task involves more rewards with some of them non-differentiable.

We use the quadrotor simulator VisFly (Li et al. (2024)) as our training environment, where the dynamics are well implemented with automatic FOG computation achieved via (Paszke et al. (2017)).A comprehensive description of observation and reward structure is presented in Table 1.

Table 1: Observations and rewards used in benchmark quadrotor tasks

| Environments | Observation | Reward Function |
|---|---|---|
| **Hovering** | state & $\hat{p}$ | $c - k_1 \|p - \hat{p}\| - k_2 \|q - \hat{q}\| - k_3 \|v\| - k_4 \|\omega\|$ (fully DIFF) |
| **Tracking** | state & next 10 $\hat{p}_{i=1\sim10}$ | $c - k_1 \|p - \hat{p}_0\| - k_2 \|q - \hat{q}\| - k_3 \|v\| - k_4 \|\omega\|$ (fully DIFF) |
| **Landing** | state & $\hat{p}$ | $-k_1 f^+\left(\|p_{xy} - \hat{p}_{xy}\|\right) + k_2 f^+\left(\|v_z - \hat{v}_z\|\right) + k_3 s$ (partially DIFF) |
| **Racing** | state & next 2 $\hat{p}_{i=1,2}$ of gates | $c - k_1 \|p - \hat{p}_0\| - k_2 \|q - \hat{q}\| - k_3 \|v\| - k_4 \|\omega\| + k_5 s$ (partially DIFF) |

$c$ represents a small constant used to ensure the agent remains alive. $k_i$ denotes constant weights for different reward contributions, with these weights being distinctly defined for each task. $s$ is a boolean variable that indicates whether the task is successfully completed, to award once at termination if it succeeds. The state comprises position ($p$), orientation ($q$), linear velocity ($v$), and angular velocity ($\omega$). $f^+(\cdot)$ denotes an increasing mapping function used to normalize the reward and $\hat{(\cdot)}$ denotes target status. DIFF is abbreviation for differentiable. All the action types are individual rotor thrusts.

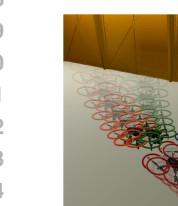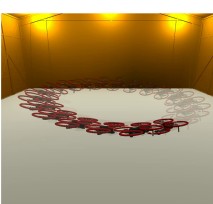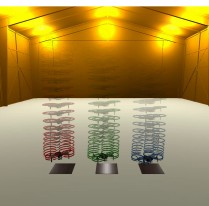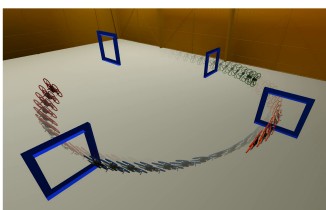

Figure 4: Quadrotor tasks (left to right): hovering, tracking, landing, and racing. We illustrate multiple drones (in different colors) simultaneously to indicate episodes from different initial states.

It is worth noting that the boolean success reward given at termination is necessary. This huge reward encourages agents to complete the mission, not wander near the target to keep obtaining the highest accumulative reward.

## 6.3 RESULTS

**Comparison with Baseline Methods**. To ensure fair comparison, we implemented SHAC and BPTT by ourselves based on available source code, and adopt PPO from stable-baselines3 (Raffin et al. (2021)) in VisFly simulator. All algorithms used parallel differentiable simulations to accelerate training. We tuned all hyperparameters to achieve optimal performance, and kept the settings consistent across all experiments as much as possible. All experiments were conducted on the same laptop with an RTX 4090 GPU and a 32-core 13th Gen Intel(R) Core(TM) i9-13900K processor, with 5 random seeds for validation of robustness. Given the different time-step metrics across the algorithms, we compare their performance in terms of wall-time as well. Figure 5 provides reward curves of all methods during training.

PPO: PPO demonstrates moderate performance across the four tasks. However, due to the lack of an analytical gradient, PPO requires more sample collections to estimate the policy gradient, making it slower in terms of time-steps. In tasks that involve fully differentiable rewards such as hovering and tracking, it achieves the lowest asymptotic reward compared to FOG-based algorithms. As expected, PPO produces smooth and acceptable learning curves, since non-differentiable rewards do not impact the ZOG used by PPO.

BPTT: BPTT exhibits similar performance to SHAC and ABPT in the first two tasks. In the Landing task, despite the reward function incorporating non-differentiable discrete scores upon success, this component has only a minor impact on the FOG computation. This is because the reward function excluding this constant, has correctly determined the gradient via backpropagation. In the Racing task, we apply learning rate decay to BPTT, SHAC, and ABPT. BPTT shows the worst performance among all algorithms, demonstrating that the iteration quickly converges to a local minimum, caused by the bias introduced by the non-differentiable part in rewards.

SHAC: Even though FOG is minimally biased in the Landing task, the curves from the five random seeds show significant fluctuations. The terminal success reward leads to high variance in the $TD(\lambda)$ formulation used to estimate N-step returns, complicating critic training. As a result, SHAC performs worse than BPTT in the Landing task. In the Racing task, the terminal value partially addresses the non-differentiable components but still performs much worse than PPO and ABPT.

Our ABPT: In all tests, our ABPT method converges to the highest rewards. It achieves the fastest convergence speeds in the first three tasks and similar convergence speed to PPO in the racing tasks. By replaying visited states as initial states, ABPT enhances sampling efficiency by exploiting corner cases. Introducing the entropy helps suppress the high variance of the discrete reward space in the landing task, contributing to greater training stability. In the racing task, ABPT also outperforms PPO with a higher converged reward. This is largely due to that the value gradient introduced by 0-step returns is unaffected by non-differentiable rewards, making ABPT an effective method to compensate for biased gradient.

**Ablation**. As shown in Figure 6, we evaluate the effectiveness of key components of our approach by removing each during training. The results show that: 1) Incorporating 0-step return clearly improves the training performance in tasks with non-differentiable rewards such as landing and racing. 2) Initializing episodes from previously visited states stored in the buffer enhances sampling efficiency, accelerating convergence. 3) In racing, the performance gain appears to stem more from entropy than from 0-step return. Actually, it is underfitting critic that deteriorates the actor training, and entropy loss helps stabilize critic training, especially when multiple critics are used. Similar to other value-based RL algorithms, convergence critically depends on the quality of critic training. 4) Removing the N-step return significantly reduce landing perfor-

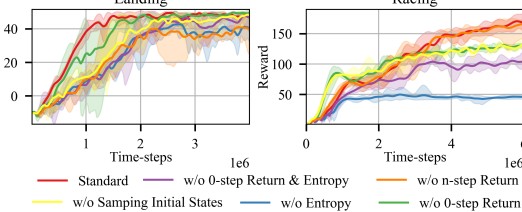

Figure 6: Ablation study: the key components of ABPT are sequentially removed in turn to evaluate each one's contribution.

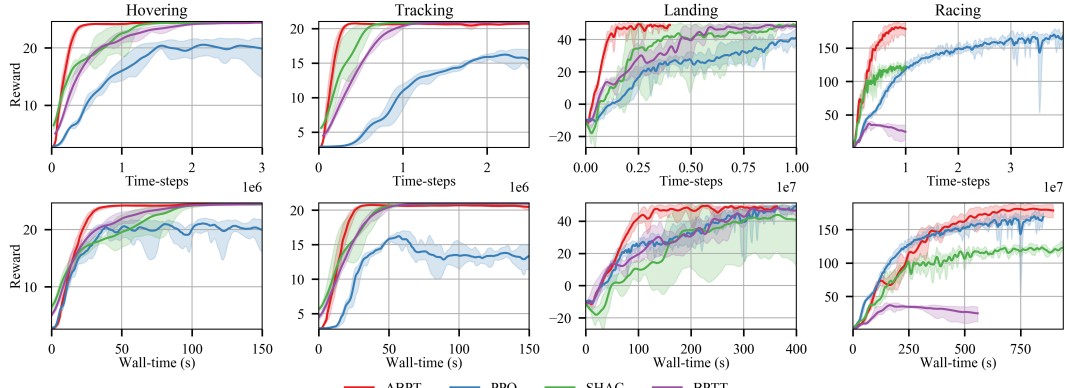

Figure 5: Training curves of PPO, SHAC, BPTT, and our ABPT in both time-step (**Top**) and wall-time (**Bottom**). Each curve is averaged over results from five random seeds, and the shaded area denotes the range of best and worst reward.

mance but has negligible impact on racing. In racing, the binary reward primarily drives the gradient, whereas in landing it serves only as an auxiliary guidance. This suggests that N-step return does not substantially contribute to mitigating biased gradients.

**Discontinuity Relaxation**. To evaluate the effectiveness of relaxation techniques in addressing non-differentiable rewards, we replace the binary reward with smooth approximations that closely resemble its behavior while using only N-step return for training. Specifically, we employ logarithmic ($-5\log(|p - \hat{p}| + 0.01)$) and exponential ($1/(|p - \hat{p}| + 0.05)$) relaxations, as shown in Figure 7. Both functions exhibit a similar trend to the original binary reward.

The objective of racing task is to pass through as many gates as possible. Since the reward scales of different relaxations are not directly comparable, the number of gates passed provides a fairer metric for performance evaluation. Although the exponential relaxation achieves performance comparable to our ABPT method, its variance is significantly higher, leading to instability. As a result, the quadrotor is more likely to become trapped in local minima. Intuitively, such relaxation encourages the quadrotor to hover near a gate to repeatedly obtain sub-optimal rewards rather than flying forward to the next gate.

**Additional results**. In real world, the increasing complexity of dynamics and variety of tasks make the optimal hyperparameters of BPTT-based algorithms to differ substantially from those reported in previous work (see Appendix C.3). We validate our trained policies through real-world experiments, as shown in Figure 1, and provide a supplementary video demonstration. The policies perform reliably, and the flights are stable. We further evaluate all the methods under different reward types ( Appendix C.1) and learning rates ( Appendix C.2), demonstrating ABPT's superior robustness.

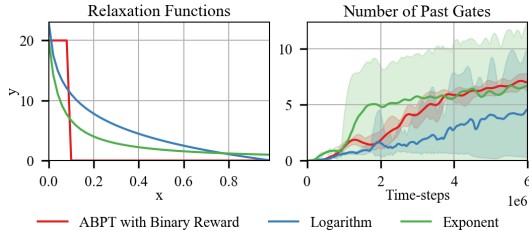

Figure 7: Training curves for the racing task using binary reward relaxations. The task is trained with SHAC after replacing the binary reward with relaxed rewards.

## 7 CONCLUSION

We present ABPT, a novel approach to train policies for quadrotor tasks robustly. It effectively addresses the challenges from the partially non-differentiable rewards associated with existing first-order gradient learning methods. We validated ABPT on four quadrotor tasks — hovering, tracking, landing, and racing — and compared them with existing learning algorithms. The results show that ABPT achieves faster and more stable training processes and converges to higher rewards across all tasks. ABPT is also robust to the learning rate and different kinds of rewards. The ablation study also shows the effectiveness of each key component of our approach.

## REPRODUCIBILITY STATEMENT

The code is released at `https://anonymous.4open.science/r/APG-E73E`. The detailed hyperparameters of all the experiments are introduced in Appendix G. Our trained policies are also deployed onboard on real-world quadrotors, please refer to the supplementary video.

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

## A  PROOF

Suppose the value function $Q_\phi$ is well trained, the accumulated reward within the horizon can be approximated as:

$$\mathcal{G}_\theta^{t:t+N} \approx \mathcal{V}_{\theta|\phi}^t - (1-d)\gamma^N \mathcal{V}_{\theta|\phi}^{t+N+1}. \tag{13}$$

Its value gradient is then given by

$$\nabla_\theta^{[q]} \mathcal{G}_\theta^{t:t+N} = \nabla_\theta \mathcal{V}_{\theta|\phi}^t - (1-d)\gamma^N \nabla_\theta \mathcal{V}_{\theta|\phi}^{t+N+1} \tag{14}$$

regardless of the differentiability of the rewards. Noting that, unlike (Xu et al. (2022)), we specifically use action-value function $Q_\phi$ to compute the value to ensure $\mathcal{G}_\theta^{t:t+N}$ is differentiable with respect to $\theta$, which makes this derivative expression meaningful, otherwise the derivative would be zero if using $V_\phi$ solely with state input. Let $\nabla_\theta \mathcal{G}_\theta^{t:t+N}$ denote the first-order gradient of the accumulated reward. The average of the two gradients can be expressed as:

$$\bar{\nabla}_\theta \mathcal{G}_\theta^{t:t+N} = \frac{1}{2} \left( \nabla_\theta^{[q]} \mathcal{G}_\theta^{t:t+N} + \nabla_\theta \mathcal{G}_\theta^{t:t+N} \right). \tag{15}$$

It is straightforward to verify that taking the derivative of (7) yields the following gradient for backpropagation:

$$\nabla_\theta \mathcal{J}_\theta = \frac{1}{|\mathcal{B}|} \sum_{i=1}^{|\mathcal{B}|} \left[ \bar{\nabla}_\theta \mathcal{G}_\theta^{t:t+N+} + (1-d)\gamma^N \nabla_\theta \mathcal{V}_{\theta|\phi}^{t+N+1} \right]$$

$$= \frac{1}{2|\mathcal{B}|} \sum_{i=1}^{|\mathcal{B}|} \left[ \underbrace{\nabla_\theta^{[q]} \mathcal{G}_\theta^{t:t+N}}_{\nabla \mathcal{J}_\theta^0} + \underbrace{\nabla_\theta \mathcal{G}_\theta^{t:t+N} + (1-d)\gamma^N \nabla_\theta \mathcal{V}_{\theta|\phi}^{t+N+1}}_{\nabla \mathcal{J}_\theta^N} \right]. \tag{16}$$

Therefore, the difference between this gradient and the gradient (4) used in (Xu et al. (2022)) is that the first-order gradients in (4) are combined with the value gradients. By leveraging this combination, our method remains effective in guiding the parameter updates toward the correct direction, when the first-order gradient is biased due to the non-differentiable rewards.

We conduct a simple experiment to assess the effectiveness of incorporating the 0-step return in addressing gradient bias. We deliberately detach parts of rewards in the hovering task (see Section 6.1) to mimic non-differentiable rewards, then backpropagate to compute gradient of network parameter. As shown in Figure 8, combining the 0-step return with the N-step return in the objective function (7) for training significantly reduces the model parameter residuals.

## B  BENCHMARK DYNAMICS

Quadrotor dynamics aligned with real-world conditions are considerably more complex than those typically assumed in simulation. The dynamics are modeled in full 6-DoF to capture the complex interactions between translational motion, rotational dynamics, aerodynamic drag, and actuator dynamics. Specifically, the state evolution is governed by:

$$
\dot{\mathbf{x}}_W = \mathbf{v}_W, \quad \dot{\mathbf{v}}_W = \frac{1}{m}\mathbf{R}_{WB}(\mathbf{f} + \mathbf{d}) + \mathbf{g},
$$
$$
\dot{\mathbf{q}} = \tfrac{1}{2}\mathbf{q} \otimes \mathbf{\Omega}, \quad \dot{\mathbf{\Omega}} = \mathbf{J}^{-1}(\boldsymbol{\eta} - \mathbf{\Omega} \times \mathbf{J}\mathbf{\Omega}),
$$

(17)

where the translational states $(\mathbf{x}_W, \mathbf{v}_W)$, orientation quaternion $\mathbf{q}$, and angular velocity $\mathbf{\Omega}$ evolve under the influence of gravity $g$, collective thrust vector $\mathbf{f}$, and drag force $\mathbf{d}$. The quaternion product is denoted by $\otimes$, and $\mathbf{R}_{WB}$ is the rotation matrix from body to world frame. $m$ and $\mathbf{J}$ respectively denote mass and inertial matrix.

The aerodynamic drag $\mathbf{d}$ is modeled as quadratic in body-frame velocity:

$$
\mathbf{d} = \tfrac{1}{2}\rho\, \mathbf{v}_B \odot \mathbf{v}_B\, \mathbf{C}_d \odot \mathbf{s},
$$

(18)

where $\rho$ is the air density, $\mathbf{C}_d$ the drag coefficients, $\mathbf{s}$ the effective cross-sectional areas, and $\mathbf{v}_B$ the velocity in the body frame. The operator $\odot$ denotes element-wise multiplication.

Under CTBR control, the action $\mathbf{a}$ consists of the collective thrust along z-axis $f$ and the desired bodyrates $(\omega_x, \omega_y, \omega_z)$. Such commands are distributed onto the four individual motors through a control allocation process:

$$
\begin{bmatrix} f_1 \\ f_2 \\ f_3 \\ f_4 \end{bmatrix} = \mathbf{M}^{-1} \begin{bmatrix} f \\ \tau_x \\ \tau_y \\ \tau_z \end{bmatrix},
$$

(19)

where $(\tau_x, \tau_y, \tau_z)$ are the body torques $\eta$ computed from the commanded bodyrates using a cascaded attitude controller. The matrix $\mathbf{M}$ denotes the allocation matrix that maps individual rotor thrusts to total thrust and body torques:

$$
\begin{bmatrix} f \\ \tau_x \\ \tau_y \\ \tau_z \end{bmatrix} = \begin{bmatrix} 1 & 1 & 1 & 1 \\ 0 & l & 0 & -l \\ -l & 0 & l & 0 \\ c_\tau & -c_\tau & c_\tau & -c_\tau \end{bmatrix} \begin{bmatrix} f_1 \\ f_2 \\ f_3 \\ f_4 \end{bmatrix},
$$

(20)

where $l$ is the arm length and $c_\tau$ is the rotor torque coefficient.

This formulation ensures that the collective thrust and commanded bodyrates are consistently mapped to the individual motor thrusts, enabling low-level execution on real quadrotors.

To account for actuator dynamics, a first-order exponential model with time constant $c$ is introduced to describe the delay between commanded and actual rotor speeds:

$$
f_i = k_2\omega_i^2 + k_1\omega_i + k_0, \quad \omega_i = \omega_i^{des} + (\omega_i' - \omega_i^{des})e^{-ct},
$$

(21)

where $\omega_i$ is the rotor speed, $\omega_i'$ and $\omega_i^{des}$ are the current and desired speeds, and $k_2, k_1, k_0$ are thrust coefficients. $f_i$ denotes the thrust along the $z$-axis of rotor $i$.

The device communication process is modeled with a one-step delay:

$$
\mathbf{a}_t = \mathbf{a}_{t-1},
$$

(22)

where $a_t$ denotes the control command applied at time $t$. This formulation captures the fact that actuators cannot instantly follow rapid changes in control inputs.

Besides, it couples a PD controller for stable bodyrate response. The actual bodyrate command is computed through:

$$\boldsymbol{\tau} = K_p^\Omega \, (\boldsymbol{\Omega}^{des} - \boldsymbol{\Omega}) \; + \; K_d^\Omega \, (\dot{\boldsymbol{\Omega}}^{des} - \dot{\boldsymbol{\Omega}}), \tag{23}$$

Then, to reduce simulation-to-reality gap, we made parameter recognition to finetune the parameters in simulation, aligning the control response as similar as possible. Such complexity makes first-order gradient computation in backpropagation particularly challenging.

## C  ADDITIONAL EXPERIMENT

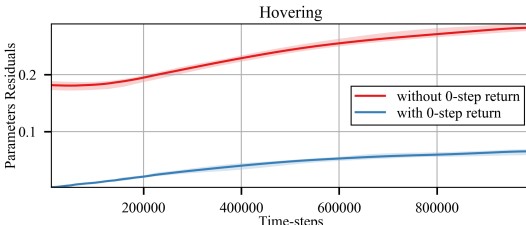

Figure 8: The curve shows the difference between the parameters trained with fully differentiable and partially differentiable rewards. We deliberately detach parts of the rewards to interrupt gradient backpropagation and retrain the policy with or without combining the 0-step return.

### C.1  REWARD ROBUSTNESS

Designing an appropriate reward function is highly challenging for real-world applications, particularly when dealing with specific requirements. Ensuring robustness to reward architecture is crucial for the training algorithms. In the racing task, we redefined the reward function by replacing Euclidean distance with approaching velocity in the reward. As shown in Figure 9, ABPT outperforms other methods with both position-based and velocity-based rewards. With fewer non-differentiable components, velocity-based rewards allow ABPT and SHAC to pass more gates per episode, while BPTT fails due to gradient issues.

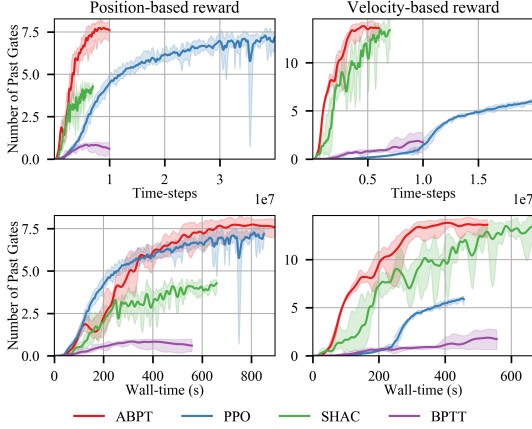

Figure 9: Training curves with different rewards: position-based rewards (**Left column**) and velocity-based rewards (**Right column**). The number of passed gates is visualized as the performance metric because of different rewards used for training.

## C.2 LEARNING RATE ROBUSTNESS

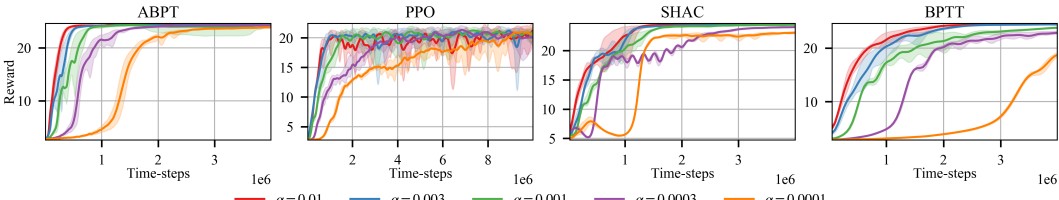

Figure 10: Training curves with different learning rates $0.01, 0.003, 0.001, 0.0003, 0.00001$. The proposed ABPT exhibits stable and fast training performance in all learning rates.

We evaluated the training performance using different learning rates. The fully differentiable hovering task is used for evaluation. As shown in Figure 10, the proposed ABPT exhibits stable and fast training performance in all learning rates. PPO has the highest variance compared to other FOG-based algorithms, as expected demonstrating that FOG is much more precise than ZOG. Increasing the learning rate yields a slight improvement on acceleration once it surpasses 0.001 for PPO and SHAC, while ABPT's convergence speed stably grows with increasing learning rate.

## C.3 HORIZON LENGTH ANALYSIS

The optimal horizon length in SHAC is typically reported as 32, but the results obtained in this work reveal a different trend. As shown in Figure 11, for the hovering task the algorithm achieves comparable final returns with horizons of 64, 96, and 128. This discrepancy can be attributed to the increased complexity of the quadrotor dynamics discussed in Appendix B. In contrast, for the landing task the set of effective horizons narrows to a single value, 96, suggesting that not only the underlying dynamics but also the task context play a crucial role in determining suitable hyperparameters.

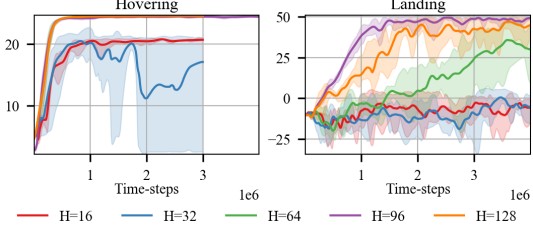

Figure 11: Training curves for hovering and landing tasks with different horizon lengths. Performance is evaluated with horizons of 16, 32, 64, 96, and 128.

## C.4 SAC COMPARISON

SAC is typically used as a baseline for value-iteration model-free algorithms and has shown strong performance across various simulation benchmarks. However, in real-world scenarios—particularly for planning six-dimensional motions in free space—the size and variance of the observation space are much greater than in simulation. This increased complexity makes it significantly more difficult to train the critic. Since the degree of critic undertraining is critical for value-iteration methods like SAC but less so for policy-iteration methods, PPO has become the most widely used algorithm for training policies deployed in practice.

To validate SAC's performance, we include it only in the hovering task (Figure 12). The results show that, in real-world applications, SAC performs much worse than PPO. Therefore, in the experiments presented in this paper, we focus our comparisons on PPO.

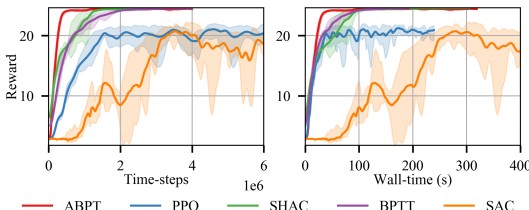

Figure 12: Training curves of hovering for baselines including SAC.

## D    INEFFICIENT SAMPLING

For convenience, the current implementation of backpropagation-through-time (BPTT) in differentiable simulation always initializes the next computation horizon from the terminal state of the previous horizon. However, this design prevents certain states from ever being sampled as initial conditions (see Figure 13), which leads to inefficient exploration of the observation space. In particular, states that are not reachable within a single horizon length cannot serve as starting points for training. This issue could be addressed by introducing an external replay buffer that records states at each step and resamples them as initial conditions, thereby improving coverage of the state space and enhancing sample efficiency. Noting that, in control task, the randomization domain could be enlarged enough to tackle such issue, but in planning task, it is usually constrained around the point of departure. Besides, regardless of randomization, the actual starting point distribution in observation space is still non-uniform, downgrading the training efficiency.

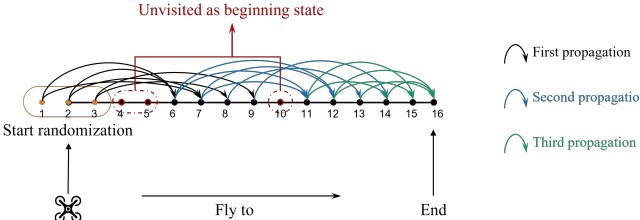

Figure 13: Illustration of the limited state coverage in the current BPTT implementation. Assuming the agent's minimal horizon length is five, it cannot end at points 4 and 5 after executing a horizon even with randomization. As a result, part of the observation space never serves as beginning of horizons, reducing sampling efficiency.

## E    DISCUSSION

We also explored incorporating $k$-step value functions ($k = 0, \ldots, N-1$) at each step within a finite horizon, following a similar approach to AOBG (Suh et al. (2022)). However, this led to significant fluctuations in the learning curves, because introducing undertrained critic results in much higher variance in training. The mixture ratio in AOBG (Suh et al. (2022)) and AGPO (Gao et al. (2024)) may be effective to handle such unstable factor. However, if for accelerating training purpose, it is impossible to directly use such method in these works because mixture ratio computation is time costly. It is worthwhile finding a much faster and simpler method for this optimization problem, similar to how PPO simplified the ideas behind TRPO.

## F    LIMITATION AND FUTURE WORK

ABPT enhances the efficiency and robustness of training processes utilizing analytical gradients, even in scenarios involving partially differentiable reward structures. However, while it significantly mitigates the gradient bias caused by non-differentiable reward components, it may still fail to fully eliminate extreme bias if the biased gradient is excessively large. Therefore, when designing reward functions, priority should be given to incorporating smooth and differentiable variables to the

greatest extent possible. In the following work, we will further explore how to adaptively mix the gradient while avoiding incurring excessive computation for mixture-ratio estimation.

# G    TRAINING HYPERPARAMETERS

Tables 2∼5 contain the parameters for the baseline experiments, Tables 6∼8 for the ablation experiments, and Table 9 for the reward robustness experiments. Noting that, Simulators like dm_control often ignore the complexity of real-world dynamics. In our case, we conducted detailed system identification and matched our simulation to actual quadrotor behavior, including communication delay, motor dynamics, aerodynamic drag, PID control, thrust modeling, and time synchronization. (see VisFly). This complexity makes optimal hyperparameters differ from original baselines.

Table 2: Hyperparameters of SHAC

|  | Hovering | Tracking | Landing | Racing |
|---|---|---|---|---|
| learning rate $\alpha$ | 0.01 | 0.01 | 0.01 | 0.002 |
| number of parallel environments $n$ | 100 | 100 | 100 | 100 |
| discount factor $\gamma$ | 0.99 | 0.99 | 0.99 | 0.99 |
| training critic steps per minibatch | 10 | 10 | 10 | 10 |
| weight decay | 0.00001 | 0.00001 | 0.00001 | 0.00001 |
| target critic update factor $\tau$ | 0.005 | 0.005 | 0.005 | 0.005 |
| decay learning rate | False | False | False | True |
| value estimation factor $\lambda$ | 0.95 | 0.95 | 0.95 | 0.95 |
| horizon length $H$ | 96 | 96 | 96 | 96 |
| Optimizer | Adam | Adam | Adam | Adam |

Table 3: Hyperparameters of PPO

|  | Hovering | Tracking | Landing | Racing |
|---|---|---|---|---|
| learning rate $\alpha$ | 0.001 | 0.0002 | 0.0005 | 0.001 |
| number of parallel environments $n$ | 100 | 100 | 100 | 100 |
| discount factor $\gamma$ | 0.99 | 0.99 | 0.99 | 0.99 |
| minibatch size | 25600 | 25600 | 25600 | 51200 |
| training critic steps per minibatch | 5 | 5 | 5 | 5 |
| weight decay | 0.00001 | 0.00001 | 0.00001 | 0.00001 |
| GAE $\lambda$ | 1 | 1 | 1 | 1 |
| Optimizer | Adam | Adam | Adam | Adam |

Table 4: Hyperparameters of BPTT

|  | Hovering | Tracking | Landing | Racing |
|---|---|---|---|---|
| learning rate $\alpha$ | 0.01 | 0.01 | 0.005 | 0.002 |
| number of parallel environments $n$ | 100 | 100 | 100 | 100 |
| discount factor $\gamma$ | 0.99 | 0.99 | 0.99 | 0.99 |
| weight decay | 0.00001 | 0.00001 | 0.00001 | 0.00001 |
| decay learning rate | False | False | False | True |
| horizon length $H$ | 256 | 256 | 256 | 512 |
| Optimizer | Adam | Adam | Adam | Adam |

Table 5: Hyperparameters of ABPT

|  | Hovering | Tracking | Landing | Racing |
|---|---|---|---|---|
| learning rate $\alpha$ | 0.01 | 0.01 | 0.01 | 0.01 |
| number of parallel environments $n$ | 100 | 100 | 100 | 100 |
| discount factor $\gamma$ | 0.99 | 0.99 | 0.99 | 0.99 |
| training critic steps per minibatch | 10 | 10 | 10 | 10 |
| weight decay | 0.00001 | 0.00001 | 0.00001 | 0.00001 |
| target critic update factor $\tau$ | 0.005 | 0.005 | 0.005 | 0.005 |
| decay learning rate | False | True | False | True |
| value estimation factor $\lambda$ | 0.95 | 0.95 | 0.95 | 0.95 |
| horizon length $H$ | 96 | 96 | 96 | 96 |
| replay buffer size | 1000000 | 1000000 | 1000000 | 50000 |
| Optimizer | Adam | Adam | Adam | Adam |

.

Table 6: Hyperparameters of ABPT no 0-step Value

|  | Hovering | Tracking | Landing | Racing |
|---|---|---|---|---|
| learning rate $\alpha$ | 0.01 | 0.01 | 0.01 | 0.01 |
| number of parallel environments $n$ | 100 | 100 | 100 | 100 |
| discount factor $\gamma$ | 0.99 | 0.99 | 0.99 | 0.99 |
| training critic steps per minibatch | 10 | 10 | 10 | 10 |
| weight decay | 0.00001 | 0.00001 | 0.00001 | 0.00001 |
| target critic update factor $\tau$ | 0.005 | 0.005 | 0.005 | 0.005 |
| decay learning rate | False | True | False | True |
| value estimation factor $\lambda$ | 0.95 | 0.95 | 0.95 | 0.95 |
| horizon length $H$ | 96 | 96 | 96 | 96 |
| replay buffer size | 1000000 | 1000000 | 1000000 | 50000 |
| Optimizer | Adam | Adam | Adam | Adam |

Table 7: Hyperparameters of ABPT no Entropy

|  | Hovering | Tracking | Landing | Racing |
|---|---|---|---|---|
| learning rate $\alpha$ | 0.01 | 0.01 | 0.002 | 0.002 |
| number of parallel environments $n$ | 100 | 100 | 100 | 100 |
| discount factor $\gamma$ | 0.99 | 0.99 | 0.99 | 0.99 |
| training critic steps per minibatch | 10 | 10 | 10 | 10 |
| weight decay | 0.00001 | 0.00001 | 0.00001 | 0.00001 |
| target critic update factor $\tau$ | 0.005 | 0.005 | 0.005 | 0.005 |
| decay learning rate | False | True | False | True |
| value estimation factor $\lambda$ | 0.95 | 0.95 | 0.95 | 0.95 |
| horizon length $H$ | 96 | 96 | 96 | 96 |
| replay buffer size | 1000000 | 1000000 | 1000000 | 50000 |
| Optimizer | Adam | Adam | Adam | Adam |

Table 8: Hyperparameters of ABPT no 0-step Value no Entropy

|  | Hovering | Tracking | Landing | Racing |
|---|---|---|---|---|
| learning rate $\alpha$ | 0.01 | 0.01 | 0.002 | 0.002 |
| number of parallel environments $n$ | 100 | 100 | 100 | 100 |
| discount factor $\gamma$ | 0.99 | 0.99 | 0.99 | 0.99 |
| training critic steps per minibatch | 10 | 10 | 10 | 10 |
| weight decay | 0.00001 | 0.00001 | 0.00001 | 0.00001 |
| target critic update factor $\tau$ | 0.005 | 0.005 | 0.005 | 0.005 |
| decay learning rate | False | True | False | True |
| value estimation factor $\lambda$ | 0.95 | 0.95 | 0.95 | 0.95 |
| horizon length $H$ | 96 | 96 | 96 | 96 |
| replay buffer size | 1000000 | 1000000 | 1000000 | 50000 |
| Optimizer | Adam | Adam | Adam | Adam |

Table 9: Hyperparameters of ABPT upon Velocity-based Reward in Racing

|  | ABPT | SHAC | BPTT | PPO |
|---|---|---|---|---|
| learning rate $\alpha$ | 0.02 | 0.02 | 0.002 | 0.0002 |
| number of parallel environments $n$ | 100 | 100 | 100 | 100 |
| discount factor $\gamma$ | 0.99 | 0.99 | 0.99 | 0.99 |
| training critic steps per minibatch | 10 | 10 | 10 | 5 |
| weight decay | 0.00001 | 0.00001 | 0.00001 | 0.00001 |
| target critic update factor $\tau$ | 0.005 | 0.005 | - | - |
| decay learning rate | True | True | True | - |
| value estimation factor $\lambda$ | 0.95 | 0.95 | - | - |
| horizon length $H$ | 96 | 96 | 512 | - |
| replay buffer size | 50000 | - | - | - |
| minibatch size | - | - | - | 51200 |
| GAE | - | - | - | 1 |
| Optimizer | Adam | Adam | Adam | Adam |

