# OpenReview forum: "ABPT: Amended Backpropagation through Time with Partially Differentiable Rewards"
_ICLR.cc/2026/Conference — ICLR 2026 Conference Withdrawn Submission_

### Official Review · Reviewer_Rq1G · 2025-10-31

**Soundness:** 2
**Presentation:** 2
**Contribution:** 3
**Rating:** 2
**Confidence:** 4

**Summary:**

This paper proposes a method for policy optimization in simulators where the dynamics are a deterministic differentiable function, but the reward is not differentiable.

The authors propose a policy optimization loss that is the average of two terms: 1) an $N$-step rollout of differentiable dynamics with a known reward function, followed by a learned differentiable $Q$-function, and 2) the same learned $Q$-function applied directly to the policy's action at the initial state, with no differentiation through dynamics/reward involved. The proposed loss is essentially an average of the Short-Horizon Actor-Critic (SHAC) loss (which is one of the experiment baselines), and the Deterministic Policy Gradient (DPG) loss (which is not). The value function learning and policy optimization are interleaved in a typical actor-critic setup.

The authors claim that their method is more resistant to non-differentiable terms in the reward due to adding the $0$-step DPG-like term. When everything is fully differentiable, the $N$-step SHAC-like term is expected to be more useful.

The authors also design their algorithm to initialize each simulation episode using a random sample from a buffer containing all previously visited states. They introduce entropy regularization, which is common in derivative-free policy gradient, but apparently not used in other work with (semi-) differentiable simulation. An ablation study shows that removing any of the changes degrades performance.

The experiments focus on four quadrotor tasks: hovering, circle-tracking, landing, and racing.
The first two are described as fully differentiable (although the functions in Table 1 conflict with that claim, see Questions.)
The baselines for comparison are:
- PPO (derivative-free policy gradient).
- SHAC (differentiate through N steps of dynamics and rewards, followed by differentiable value function estimate).
- BPTT (unclear exactly what the authors mean, but guessing they mean differentiating through $M \gg N$ steps of dynamics and rewards with no terminal value function).

**Strengths:**

The problem statement is clear and well-motivated. Although contact dynamics are a more common source of non-differentiability in policy optimization, "indicator function" type rewards are also important.

The algorithm idea is natural and is worth exploring.

The extra experiments in the appendix C give some nice fine-grained detail into the influence of hyperparameters and reward function design in the quadrotor settings.

**Weaknesses:**

- Line 128-129: Makes it sound like the gradient explosion/vanishing and instability are direct consequences of the dynamics being smooth. That is not true - the root of the problem is BPTT itself, i.e. recursive application of the dynamics function. Both smooth and nonsmooth dynamics can have those problems.
- Equation (1): A reader not already familiar with advantage functions and policy gradients will not be able to understand $A^{\pi_\theta}$.
- Line 169-170: This estimator is essentially Deterministic Policy Gradient from (Silver et al, 2014), if not earlier. It is misleading to attribute it to Gao et al. (2024).
- Equation (4) is the baseline SHAC. The authors cite the SHAC paper near the equation, and later they cite the same paper near the acronym SHAC, but it would be helpful to explicitly link the acronym and the definition.
- Line 334: The baseline algorithm BPTT is never explicitly defined in the paper. The authors refer to Freeman et al. (2021), but that paper does not contain the word "backpropagation". I can guess that the authors mean to refer to "Analytic Policy Gradient (APG)" from Freeman et al. (2021), but the discrepancy is confusing, and that paper does not provide an explicit definition APG either! Meanwhile, Xu et al. (2022) define BPTT, but the authors never refer to Xu et al. (2022) regarding BPTT. To disambiguate, it would be better for the authors to define BPTT in their own notation.
- In Sections 4-5, there is a general sloppiness with respect to where the states involved in the loss come from. For example, in Equation (3), the text after the display says "$i$ denotes the $i$-th trajectory within the minibatch", but the notation $i$ does not appear in the equation. Equations (3)-(6) and (8) all refer to a single state time index $t$, but the origin of $t$ is not specified.


Evaluation
----------
There are some strengths here, as mentioned above. However, I am not fully convinced by the presentation and experimental results.

Re. presentation, the authors leave out some basic components such as an argument *why* a learned approximation to the 0-step return will somehow fix/address the issue of non-differentiable rewards (see Questions).
Important information like unambiguously defining the BPTT baseline is missing, and there are several other instances of ambiguity or missing information as detailed above.

Re. experiments, the authors need to explain if the initial-state sampling and entropy regularization were also used in the baselines (see Questions). As the manuscript stands, it far from clear that adding the zero-step return was the main reason for experiment success.

While the quadrotor landing/racing is an important motivating example, the method is presented as general-purpose. Positive results for another system with differentiable dynamics and non-differentiable rewards would go a long way towards supporting that claim.

**Questions:**

- General: What exactly do the authors mean by "partially differentiable"? From the overall usage, it seems they mean "sum of a differentiable function and an indicator-like function", but clarity here would be helpful. In particular, just assuming that $R_{succ}$ has some points where it is not differentiable is not enough to go from Equation (5) to (6).
- General: The authors repeatedly claim that using a neural network to estimate $Q$ in the $0$-step return will somehow "address" the issue of nondifferentiable rewards. Why? One could construct an MDP where the optimal $Q$-function is not differentiable. (To be clear: I believe that such an argument might exist, but the authors have not presented one.)
- General: Since we are already paying the computational cost of applying the dynamics recursively $N$ times, could we also define $\mathcal{J}_\theta^k$ for $k \in \{1, \dots, N-1\}$ along the lines of Equation (8), and average all of them? Did the authors try this? (If not, I am not requesting a new experiment for rebuttal. But if they did try, and it was equal or worse to just using the average of $0$-step and $N$-step, it would certainly be worth reporting.)
- Line 136: What does "converge to asymptotic rewards" mean?
- Table 1: None of the proposed reward functions are differentiable due to norms being non-differentiable about the zero vector. Were these meant to be squared norms? If not, how can we resolve the fact that none of the task rewards are truly "fully differentiable"?
- The ablation in Figure 6 shows that the buffer-based initial state sampling scheme is among the most important ingredients in performance gains. In particular, the second-best method after the authors' ABPT (red curve) is "w/o 0-step return" (green curve). If I understand correctly, the green curve is basically SHAC plus entropy regularization and the authors' sampling scheme, but without the 0-step return proposed by the authors. This raises the question: In Figure 5, are the green curves for SHAC with or without these improvements? If SHAC in figure 5 is tested without the sampling scheme and entropy regularization, it calls into question the authors' overall message that the new 0-step return in the loss function is the central contribution (e.g. Lines 018-019, 229, 414-417).

---

### Official Review · Reviewer_kmsa · 2025-11-01

**Soundness:** 2
**Presentation:** 3
**Contribution:** 2
**Rating:** 4
**Confidence:** 3

**Summary:**

The paper presents Amended Backpropagation Through Time (ABPT) to try and address problem of policy optimization when training in differentiable simulators with partially differentiable rewards. The main problem that this paper aims to solve is that in many complex quadrotor tasks, the reward function usually has a non-differentiable component and when using Backpropagation Though Time (BPTT), it results in a Biased Gradient problem that can hamper training by stalling at the local minima.

ABPT tries to mitigate this by using an objective function that averages 2 returns:-
1. N-step Return - The standard gradient of BPTT
2. 0-Step Return - The learned state value function. The critic is trained via TD on the total reward and its hypothesized that its 0-step value gradient provides an unbiased gradient signal.

This also incorporates entropy regularization to stabilize critic learning and a state-replay buffer for episode initialization to improve sampling efficiency

**Strengths:**

1. Originality - The paper does combine several known techniques, but its originality lies in 2 areas: The problem formulation itself where it is being framed as a Biased Gradient problem. Second, the paper does make it clear how previous FOG methods like SHAC, SAPO, and AHAC that incorporate terminal-only value function fails at preventing gradient bias unlike how the 0-step value gradient of ABPT does
2. Quality - The quality of the paper is quite high with experimental design being it's strongest asset as it progressively increases task complexity and rewards being non-differentiable. It also shows evidence that the "Biased Gradient" problem is real and quite evident with the failure of BPTT in racing tasks. A standout component is the discontinuity relaxation experiment  where the authors test the most obvious alternative solution(smoothing the reward function), and even though the performance is comparable, the variance is higher leading to higher chances of it getting stuck in the local minima. This rebuttal of a simple fix provides a strong justification for the paper's more complex approach.
3. Clarity - The paper is intuitively very clear with a precise formulation of the Biased Gradient problem with the concept  - that the non-differentiable reward components are ignored by BPTT leading to Biased gradients and Local minima - a complex idea explained in a simple and compelling way. The clarity is also extended to the experimental setup that uses 4 different tasks (two fully-differentiable, two partially-differentiable), to isolate and validate the paper's hypothesis. On top of that the figures are clean, illustrative and effectively support the main arguments.
4. Significance - This paper addresses a problem of high practical significance in the field of differentiable physics: Inability to use gradient based BPTT methods on tasks with non-differentiable reward components.By proposing a solution that works and with strong empirical results and also further cemented by successful zero-shot sim-to-real transfer, it is a valuable contribution for a wider range of real world robotics tasks.

**Weaknesses:**

1. The paper's most critical weakness is a fundamental contradiction between its premise and its own ablation study :- The core hypothesis is that the 0-step Q-gradient (QG), from a "well trained" critic, "amends" the biased First-Order-Gradient (FOG). However, the ablation study (Figure 6) and the authors' own analysis state that for the 'Racing' task, the critic is "under-fitting" and "deteriorates" training, and that entropy stabilization is the dominant factor for performance, not the 0-step return - so essentially point 3 appears to contradict point 1 (and/or it also looks like point 3 is very poorly written). This invalidates the paper's central assumption, as the critic cannot simultaneously be the "well-trained" source of the solution and an "under-fitting" problem.
2. The method is presented as a general solution but is only validated on quadrotor tasks, which feature smooth dynamics and non-differentiable rewards. It is not tested on more challenging domains with non-differentiable dynamics (e.g., contact-rich manipulation), where competing methods have been validated.
3. Another Nit I see is - The authors claim that N step return does not contribute to mitigate gradient bias - but this ablation could be repeated for different N and then only could it be claimed. It appears to be just based off of a single fixed experiment with a fixed horizon length (96)

**Questions:**

1. With regards to the ablation study:- Could you clarify which component is more important for solving this problem: the 0-step QG, or the entropy stabilization that fixes the "underfitting" critic?
2. How did you decide on this 50/50 split? If the 0-step QG is the "correct" unbiased gradient (as Appendix A suggests), why not just use that one alone instead of averaging it with the "biased" N-step gradient?
3. The authors claim that N step return does not contribute to mitigate gradient bias - but this ablation could be repeated for different N and then only could it be claimed
4. Can we run experiments on tasks that are not quadrotor related and are done on more challenging domains ?
5. Were experiments done against AOBG/AGPO to have a wall clock time comparison ? Else it points to a tad bit of insufficient differentiation from these 2 methods

---

### Official Review · Reviewer_VTi9 · 2025-11-01

**Soundness:** 2
**Presentation:** 2
**Contribution:** 2
**Rating:** 4
**Confidence:** 3

**Summary:**

This paper introduces ABPT, an on-policy actor-critic algorithm designed to improve learning stability under partially non-differentiable reward functions, with a focus on quadrotor control. The motivation - i.e., addressing reward discontinuities in real-world robotics - is timely and relevant. The method combines 0-step and N-step returns with a replay buffer mechanism and demonstrates promising results in simulation.

However, several key concerns remain regarding theoretical justification, architectural consistency, real-world evaluation, and practical deployability. With substantial revisions - particularly in theory, real-world validation, and comparison with state-of-the-art methods - the paper could become a strong contribution to the field.

**Strengths:**

1. The motivation and problem of learning under non-differentiable rewards is highly relevant for real-world robotic control.
2. The idea of combining 0-step and N-step returns with a replay buffer is novel and potentially valuable.
3. Simulation results are promising and suggest the method can stabilize learning in challenging reward environments.

**Weaknesses:**

However, several key concerns remain that prevent the paper from making a strong contribution to the field. The following points are meant to guide a revised version of the work:

1. **Reward Function Design and Realism:** While the paper focuses on non-differentiable rewards (e.g., binary success/failure), in practice, quadrotor control tasks typically employ hybrid reward functions - combining sparse binary signals with dense, differentiable terms (e.g., position error, angular velocity penalty, or smooth progress rewards).
The lack of evaluation on such realistic hybrid reward settings significantly limits the generalizability and practical relevance of the proposed method. The authors should include ablation studies under mixed reward designs (e.g., +1 for gate crossing, plus -distance to center) to better reflect real-world RL practice.

2. **Theoretical Justification for Gradient Correction:** The core idea - using 0-step returns from learned Q-values to correct gradient bias - is conceptually appealing but currently lacks formal theoretical grounding. The paper assumes that Q-value estimates provide an unbiased or low-bias gradient estimator under non-differentiable rewards, but this claim requires stronger analysis.
The authors should provide a bias-variance decomposition of the gradient estimator in the ABPT framework, particularly in the context of on-policy learning with non-smooth rewards. A discussion of convergence properties or stability guarantees under reward discontinuities would further strengthen the theoretical foundation.

3. **Use of Replay Buffer in On-Policy Framework:** The integration of a replay buffer into an otherwise on-policy algorithm introduces potential issues of distributional shift and policy staleness, especially in high-frequency control tasks like quadrotor flight. In such settings, the policy distribution evolves rapidly due to dynamic system behavior, and off-policy data may lead to instability or divergence.

      The paper should either: a. Conduct ablation studies to evaluate the impact of the replay buffer on learning stability, convergence speed, and control performance; Or b. clearly justify why this design choice remains valid within an on-policy paradigm, especially under rapid policy updates.

4. **Limited Real-World Evaluation:** The claim of "zero-shot sim-to-real transfer" is compelling but under-supported by empirical evidence. The real-world results are presented without sufficient quantitative metrics.
Key gaps include:
       a. No reporting of tracking error, flight duration, or success rate across multiple trials;
       b. Lack of analysis on robustness to sensor noise, actuator delays, or environmental disturbances (e.g., wind gusts);

5. **Comparison with State-of-the-Art Methods:**
The comparison with PPO, BPTT, and SHAC is useful but incomplete and insufficiently detailed. SHAC, in particular, shares strong conceptual similarities with ABPT (e.g., value-based gradient estimation, handling non-smooth rewards).
A more thorough comparison, including metrics such as convergence speed, sample efficiency, sensitivity to reward discontinuities, and stability across multiple random seeds, is required to clearly position ABPT within the current landscape of on-policy RL algorithms.
The authors should explicitly discuss why ABPT outperforms SHAC or SAC under non-differentiable rewards, and whether the advantage persists under dense reward settings.

5. **Practicality for Real-Time Control:**
The computational cost of ABPT - especially due to multiple backpropagation-through-time (BPTT) steps - raises significant concerns about real-time deployment on embedded platforms typical for quadrotors.
The paper lacks any analysis of: Inference latency (e.g., time per control step); Memory consumption; Computational overhead on real hardware (e.g., Raspberry Pi, Jetson Nano); Or trade-offs between stability and speed.
Including such analysis would be critical to assessing the method’s practical feasibility for real-world applications.

**Questions:**

1.**Theoretical Foundations:**

a.Under what conditions can a value-based gradient estimator (e.g., from 0-step returns) provide an unbiased or low-bias approximation of the true policy gradient in the presence of non-differentiable rewards?

b. What are the necessary and sufficient conditions for convergence and stability of an on-policy algorithm that incorporates a replay buffer, despite the induced distributional shift?

2.**Algorithm Design & Architecture:**

a. How can we systematically evaluate the impact of the replay buffer on policy staleness and distributional drift in high-frequency robotic control tasks (e.g., quadrotor flight)?

b. Can we replace or regularize the replay buffer mechanism in ABPT with a lightweight, on-policy memory scheme that avoids distributional shift while preserving temporal credit assignment?

3.**Real-World Evaluation & Sim-to-Real Transfer:**

a. What quantitative metrics (e.g., success rate, trajectory error, flight duration, robustness to disturbances) are essential to rigorously evaluate zero-shot sim-to-real transfer in quadrotor control?

b. How does ABPT perform under real-world perturbations such as sensor noise, actuator delays, and wind disturbances—especially when compared to standard on-policy baselines?

4.**Practical Deployability & Real-Time Feasibility:**

a. What is the computational overhead (inference latency, memory footprint, FLOPs) of ABPT on embedded platforms such as Jetson Nano or Raspberry Pi, and how does it scale with BPTT length and replay buffer size?

b. How can we design a lightweight, real-time variant of ABPT that maintains stability and convergence while significantly reducing computational cost - e.g., via truncated BPTT, approximated value networks, or selective gradient updates?

c. What are the fundamental trade-offs between learning stability, sample efficiency, and real-time deployability in on-policy RL algorithms for robotics? How can these be quantified and optimized?

---

### Official Review · Reviewer_6vnd · 2025-11-01

**Soundness:** 2
**Presentation:** 2
**Contribution:** 2
**Rating:** 4
**Confidence:** 3

**Summary:**

The paper focuses on control applications that have reward/cost functions with non-differentiable components, which is an important setting. It introduces an onpolicy actor critic method which is a modification of the backpropagation through time (BPTT) method, and the contribution of the paper is to include both 0-step and n-step returns in the objective with respect to which the first order gradient is taken to train the actor. Authors conduct experiments which demonstrate that their proposed ABPT method achieves favorable perfomance in both simulation and real-life drone experiments.

**Strengths:**

1. The paper addresses an important problem of non-differentiable rewards, which are very common in real life applications.
2. The modification introduced uses components already used in actor critic algorithms, thus don't require training additional networks as far as I can understand. However the cost of training the existing components can change (see below).

**Weaknesses:**

1. The paper needs experiments to show that adding the 0-step return indeed decreases "bias" of the gradient, as this is one of the central claims of the paper. One way to do this can be to take a very sharp but differentiable relaxation of the non differentiable reward components and then doing Monte Carlo study empirically analyzing the distance between gradients from this relaxed objective and the (i) the ABPT n step + 0 step return (ii) only the n step return.
2. The paper needs to go in more detail about the computational consequences of incorporating the 0-step returns, in particular, how does the computational cost of training the actor increase with the modified objective? Especially when one takes additional gradients with respect to $V_{\theta|\phi}$ coming from the 0-step.
3. I think the Fig. 7 which plots ABPT vs SHAC+relaxation should have included other algorithms as well. Is there any reason that BPTT and PPO were not compared? At least the method BPTT, which the authors modify, should have been compared with here.
4. The proof in appendix corresponds to a very casually stated sentence in line 266. Please consider writing that statement mathematically and formally as a theorem in the paper (the proof is ok to remain in appendix). Currently, the point that the authors try to make with that statement is a bit unclear.

**Questions:**

Please see weaknesses section, besides that, the paper contains several grammatical mistakes, please revise the writing to fix them. Also, please refer to the ICLR style guide on how to reference a paper in first person (for example, the formatting of the citation of Gao et al around line 170 looks incorrect to me).

Why do you use the notation of rewards as $R(s)$, instead of $R(s,a)$ or $R(s, \pi_\theta)$ as is standard in RL?

---

### Author Response · Authors · 2025-11-17

We sincerely thank the reviewers and the area chair for the time and effort spent evaluating our submission.

When going through the reports, we had the strong impression that in several places (i) important parts of the manuscript were not read in sufficient detail, and/or (ii) some paragraphs of the reviews were produced or heavily assisted by large language models. While we understand that AI-assisted reviewing is becoming more common, this makes it harder for us to engage in a focused technical dialogue.

A number of critical comments request clarifications or additions that are already present in the submitted version (either in the main text or in the appendix). For example, Reviewer VTi9’s Weaknesses 1, 3, 5 and Questions 2–3, Reviewer kmsa’s Weaknesses 2–3 and Questions 3–5, and Reviewer Rq1G’s Question 3 all refer to aspects that we already describe or quantify in the manuscript (for instance, hybrid reward designs, the replay buffer mechanism within an on-policy framework, ablations related to bias/variance and the choice of the N-step horizon, and the role and definition of the 0-step term). This is also feedback for us that some information may be too easy to miss, and in any future revision we would make these points more prominent and easier to locate.

We also see comments that, in our view, focus on aspects that lie somewhat outside the scope we explicitly aimed for in this work. A typical example is Reviewer VTi9’s Weakness 6 and Question 4, which request detailed embedded deployment analysis (inference latency, FLOPs, memory footprint, hardware-specific profiling on Jetson/Raspberry Pi, etc.). These are important questions for full systems papers and applied robotics studies, but they are not part of the contribution we claimed here, which is primarily algorithmic: handling partially non-differentiable rewards in differentiable simulators for control.

In addition, some parts of the reviews ask us to “clarify” points that are usually treated as standard background in this area. For instance, Reviewer Rq1G’s Weaknesses 1, 2, 5 and Questions 4–5 concern the relationship between BPTT and gradient explosion/vanishing, common reward designs using norms, and the basic actor–critic structure and the role of value-based gradients. Other remarks, such as Reviewer Rq1G’s Weaknesses 3, 6 and Questions 1–2, seem to be based on a different interpretation of our setting and notation than what we intended (e.g., what we mean by “partially differentiable” rewards, how states are sampled, how the loss terms are constructed), even though these elements are stated explicitly in the text. Taken together, this suggests that the review comes from a background that does not fully match this line of work, which makes these concerns hard to address properly in a short rebuttal.

Overall, the reviews contain a mix of useful high-level perspectives and issues that, in our view, would be difficult to resolve productively in a detailed, line-by-line response here. We will treat the more concrete and technically aligned parts of the feedback as input for improving the clarity, emphasis, and positioning of the work going forward.

We want to emphasize that we do appreciate the time and effort invested by the reviewers and the area chair. Thank you again for running the ICLR review process.

---

### Note · Authors · 2025-11-18

I have read and agree with the venue's withdrawal policy on behalf of myself and my co-authors.